# Factors Influencing Technology Integration in the Curriculum for Taiwanese Health Profession Educators: A Mixed-Methods Study

**DOI:** 10.3390/ijerph16142602

**Published:** 2019-07-22

**Authors:** Hsiao-Jung Chen, Li-Ling Liao, Yu-Che Chang, Chung-Chih Hung, Li-Chun Chang

**Affiliations:** 1School of Nursing, Chang Gung University of Science and Technology, Taoyuan City 33303, Taiwan; 2Department of Health Management, I-Shou University, Kaohsiung City 82445, Taiwan; 3Chang Gung Medical Education Research Center, CGMERC, Chang Gung Memorial Hospital, Linkou, Taoyuan City 333, Taiwan; 4Chang Gung University College of Medicine, Taoyuan City 33303, Taiwan; 5Department of Emergency Medicine, Chang Gung Memorial Hospital, Linkou, Taoyuan City 333, Taiwan; 6Department of Laboratory Medicine, Taipei Hospital, Ministry of Health and Welfare, New Taipei City 24213, Taiwan; 7Graduate Institute of Technological and Vocational Education, National Taipei University of Technology, Taipei City 10608, Taiwan; 8Department of Nursing, Chang Gung Memorial Hospital, Linkou, Taoyuan City 333, Taiwan

**Keywords:** technological pedagogical content knowledge (TPCK/TPACK), technology-enhanced learning (TEL), medical teacher, professional development

## Abstract

In this study, we explored the considerations and the influencing factors for the technological integration of educational curricula based on the technological pedagogical content knowledge (TPCK) framework for health profession educators (HPEs). A mixed methodology was used that included semi-structured interviews with 15 HPEs and an online TPCK survey with a randomly selected sample of 319 HPEs from 217 Taiwanese universities. Five themes emerged, namely, supplementing traditional teaching methods, improving immediate educator–student interactions, tracking the learning process and improving the record-keeping, keeping up with technological trends, and advancing professional learning objectives for different student needs. The presence of pre-existing platforms and inspiration from colleagues and students were facilitators, whereas the risk of technological problems and the need to invest extra time into the preparation process were impediments to technology integration in the curriculum. Of the online respondents (n = 210), 64.2% have integrated at least one technological method into their curriculum. The logistic regression model revealed that gender, prior knowledge regarding how to incorporate technology with teaching, high mean TPCK scores, and relevant school policies were significant predictors of technology integration in the curriculum. Based on these results, recommendations for development in the health profession included efforts to equip HPEs with TPCK, in order to integrate technology into the curriculum effectively.

## 1. Introduction

In the 21st century, technology has been a primary driver of change in the lives of humans. Technology has even become a significant part of the school environment; in fact, educators use it every day. In order for tomorrow’s health professionals to practice at the peak of their abilities, medical education must be attractive and innovative, moving beyond an entrenched curriculum and pedagogical approaches, to meet the needs and best develop the skills of a new generation of students [1,2]. Undoubtedly, preparing students for future challenges, in highly complex patient-care environments, invariably involves engaging students in collaborative work and real-world problem-solving through effective exploitation of information and communications technologies (ICT) [3]. However, limited data are available regarding technology integration in the curriculum of health profession educators. 

Technology plays an important role in helping students become active learners in the 21st century. Interaction and reflection support the student–educator exchange; and technology has created new and diverse methods for conducting these processes, supporting the learner-centered education in multiple ways [4]. Technology integration is the use of technology tools in general content areas in education in order to allow students to apply computer and technology skills to learning and problem-solving [5]. The integration of technology in curricula can involve web-based technologies, mobile devices and apps, computers, tablets, and multimedia, and may even include activities that are mediated entirely in the digital form or that employ a blend of digital activities [6]. Today’s medical and nursing students favor creative, active learning environments that integrate purposeful technology use [7,8]. Educational projects for health profession students demonstrate the wide variety of ways in which technology can enhance the student experience and ultimately improve patient care [9]. Observations and data collected from a technologically integrated curriculum could help educators determine whether there is a correlation between technological integration and students’ learning motivations and preferences [10]. However, there are still some issues; such as the high cost of implementing and maintaining technology, safety for students and educators (bullying), problems with infrastructure, physical injury (repetitive strain injury), fairness, and equality [11,12,13]. 

To effectively use technology to support learner-centered education, educators should have appropriate pedagogical, technological, and content-based knowledge, and should also understand the interaction between these categories of knowledge. For a variety of reasons—including, for example, a lack of time, technological skills, resources, and stakeholder demands [14], as well as the presence of competing work demands [15]—health profession educators struggle to effectively implement technology in their curriculum. These educators may also lack motivation to integrate technology into medical education due the limited or complete lack of support from their institutions [16]. Accordingly, it is clear that multiple variables affect technology integration. Literature on the subject has presented several models in which variables classified at the school and educator levels are explained in relation to one another. Variables at the educator level include gender, technological competence, subject competence, attitude, and technological pedagogical content knowledge (TPCK) experience [17,18], while a variable at the school level is executive enforcement, which has been found to be a significant factor related to an educator’s implementation of technological integration. For a comprehensive understanding the factors related to technological integration for health profession educators, mixed methods research, using quantitative and qualitative approaches in tandem, provides a better understanding of research problems than using one approach alone [19].

Most of the literature on barriers to technology integration has been anchored in the theoretical framework provided by Ertmer [20], which classified technology integration barriers into first- and second-order barriers. First-order barriers are a lack of access to computers and software, insufficient time to plan instruction, and inadequate technical and administrative support. Second-order barriers are beliefs about teaching, beliefs about computers, established classroom practices, and unwillingness to change. In a recent study, it seems that new barriers to technology integration (believing that the top unit of an institution at which educators’ work decides whether to engage in the process of technology integration, and accepting that there are several barriers to technology integration at all times) have been to found to emerge and other barriers (lack of technology resources) have been found to decrease [21]. 

## 2. Materials and Methods

A mixed method was used. Initially, a qualitative phase was used to collect the influencing factors of educators’ adoption of technology into a curriculum, and then a quantitative phase was used to incorporate the results of qualitative data in the TPCK survey as influencing factors. The focus of the qualitative research was on the process, that is, how exactly TPCK is taught. In the development of the TPCK process, the HPEs’ considerations regarding the process, what their attitudes are, how they have changed, what problems they have encountered, what they are confused about, what their feedback is, what their final results are, etc. Since there is no research on HPEs’ TPCK, this mixed research was designed using qualitative research and then quantitative research. This can help to clarify whether HPEs have TPCK and its’ influencing factors. Finally, we tested the three models to validate the findings of the mixed data.

### 2.1. Theoretical Framework

The “technological pedagogical content knowledge” (TPCK or TPACK) of educators has been recognized as the most influential factor for the successful integration of technology into an educator’s instruction [22]. TPACK refers to the “body of knowledge that teachers now need for teaching with and about technology in their assigned subject areas and grade levels” [23].

At the heart of the TPCK framework is the complex interplay of three primary forms of knowledge: content knowledge (CK), pedagogical knowledge (PK), and technology knowledge (TK). Content knowledge (CK) is knowledge about the actual subject matter that is to be learned or taught. Pedagogical knowledge (PK) is deep knowledge about the processes and practices or methods of teaching and learning and how it encompasses (among other things) overall educational purposes, values, and aims. Technology knowledge (TK) is knowledge about standard technologies, such as books, chalk, and blackboards, as well as more advanced technologies, such as the Internet and digital video. The key to successful teaching is that educators can effectively integrate the understanding of pedagogical content knowledge (PCK). Educators use the teaching process and teaching methods (pedagogical knowledge, PK) to transform the content knowledge (CK) into the content that students can understand and learn. At the same time, students use their own cognitive levels, motivations, and interests to transform, adjust, and characterize the subject content [24]. Subsequently, Mishra and Koehler [25] have added the knowledge of technology (TK) into PCK by emphasizing the connections, interactions, affordances, and constraints between and among CK, PK, and TK. There are three ways that these forms of knowledge intersect: pedagogical content knowledge (PCK), technological content knowledge (TCK), and technological pedagogical knowledge (TPK) (Figure 1). Taken together, they intersect to become technological pedagogical content knowledge (TPCK or TPACK) as a conceptual framework to describe the knowledge base educators need to teach effectively with technology. Quantitative studies using structural equation modeling have found that six aspects of the TPACK framework—namely, TK, CK, PK, TCK, PCK, and TPK— are significant contributors to educators’ self-reported efficacy for TPACK [26,27]. 

Differences in the understanding of technical content and the application of pedagogical strategies were found among educators in different professions [28]. However, the results of a study that reviewed the empirical TPACK studies, published in previous journals from 2002 to 2011, revealed that university or college educators were the least frequently used research samples in TPCK studies, and that health profession educators (HPEs) were never the subjects of TPCK studies. 

As discussed, integrating technology into the learning environment is a complex process. Recognizing and explaining the variables affecting HPEs’ technology use may improve our understanding of the process of technological integration. This study examined the following three research questions:What were the HPEs’ considerations of technological integration into curriculum design?What barriers and/or facilitators exist with regard to the use of technological methods in their curriculum?What factors influence technological integration among HPEs?

### 2.2. Participants and Sampling Method

#### 2.2.1. Qualitative Interview

A multi-institutional qualitative study was conducted using purposive and snowball sampling methods. From a list compiled by the Ministry of Science and Technology Medical Education Project, we recruited two educators, via email, who had completed a technology integration program. After interviewing these two educators, we invited them to introduce other educators with similar experiences as potential interviewees. Using this snowball sampling method, we were able to conduct face-to-face interviews with 15 HPEs. After 15 interviews, the data became saturated such that no new meaning units could be extracted from the interview texts. All study procedures were approved by the Chang Gung Medical Foundation Institutional Review Board (No. 104-9275B).

#### 2.2.2. Online Questionnaire Survey

In 2016, a total of 4031 HPEs in 217 universities/colleges were compiled from the official website of faculty estimation in the Ministry of Education of the Republic of China. The HPEs represented a variety of health profession fields, including nursing, medicine, pharmacy, nutrition, long-term care, medical laboratory science, and radiology. The sample included educators who teach health profession students in public and private universities or colleges. 

The sample size (*n* = 368) was calculated based on the power of 0.8 and an *α* of 0.05, and *σ* = 22. The sample size for each subgroup (educators within each specialty) within the total population was calculated, and then for an estimated response rate of 30%, as mentioned in the previous studies [29], a total of 987 HPEs were taken independently from each department of the universities and colleges using systematic random sampling. 

### 2.3. Data Collection

Before the qualitative interviews, the researcher explained the purpose of the interview to the selected educators and obtained their consent. Each interview lasted 60 to 90 min. Interviews were audiotaped and transcribed by the investigator. 

The questionnaires were administered through Google forms, with the link to the questionnaire sent through emails to the selected HPEs in February 2017. Completion and submission of the questionnaire by the respondents implied consent. Two reminder emails were circulated; the first was sent one week later and the second was sent three weeks later. Participants were asked to take approximately 10–15 min to complete the questionnaire and to do so only once. We analyzed all responses received on or before May 30, 2017. All participants were rewarded with a coupon as a gift after completing the interview and questionnaire. 

### 2.4. Measures

#### 2.4.1. Interview Questions

Individual in depth interviews with HPEs were conducted using the following semi-structured questions, which were based on previous studies:What kinds of technological tools/device have you adopted in your class?Why did you consider or not consider integrating technology into teaching?What are the pros and cons of integrating technology into your teaching?Did any difficulties arise when you integrated technology into teaching?

#### 2.4.2. Online TPCK Questionnaire

The online TPCK survey includes two measures: (i) a TPCK instrument and (ii) demographic and professional variables. TPCK was measured with a modified version based on the work of Chai, Ng, Li, Hong, and Koh [26]. The TPCK instrument consisted of 32 items with responses in the form of a seven-point Likert scale from “disagree very much” (1) to “agree very much” (7). The content validity index (CVI) was 0.85, and the internal consistency of the Cronbach’s α coefficients ranged from 0.70 to 0.85 for subscales and total items in this study. Demographic and professional variables were based on the qualitative finding including gender, profession type, age, academic degree, years of teaching experience, school policy of integrating technology into the curriculum, scale of technology integration, and prior knowledge regarding how to incorporate technology with teaching. 

### 2.5. Data Analysis

#### 2.5.1. Qualitative Data Analysis

The data processing and analysis methods used in this study can be divided into two parts: qualitative data analysis and quantitative data analysis. Content analysis was used to analyze educators’ interviews [30]. All the transcribed textual data were broken down by sentences as the unit of analysis so that syntactical differences, rather than semantic differences, were used to identify the textual units. The researchers first coded the transcriptions independently and then sorted the codes to identify common themes in the data using Burnard’s [31] thematic analysis framework. Furthermore, the researchers adopted an inductive approach to analyze the data, making use of a “constant comparison” method, and related open and axial coding techniques in which the emerging concepts were firmly grounded in the collected data. At the end of the analysis, five themes regarding how technology has been used in the curriculum development were strongly supported by the narrative data. These themes were deemed to be of equal importance to the participants and will be discussed in subsequent sections. 

#### 2.5.2. Quantitative Data Analysis

The online data regarding the questionnaire were exported into Excel files(Microsoft Corp. Released 2010) and then into SPSS, version 21 (IBM Corp. Released 2012.) for descriptive and inferential analyses. The descriptive analysis concerned the mean values, standard deviation, and percentages. The main outcome (dependent) variable was technology integration (yes/no). Multivariable logistic regression models, providing odds ratios (ORs) and 95% confidence intervals (95% CIs), were made to test the associations of influencing factors, TPCK, and technology integration.

## 3. Results

### 3.1. Demographic and Professional Data of Participants

The characteristics of participants in the qualitative and quantitative phases are shown in Table 1. Three hundred and nineteen valid responses were obtained (32.3% response rate). Most of the participants were between 41 and 50 years old (interview: 60%, survey: 48.90%), female (interview: 60%, survey: 63.32%), possessed doctoral degrees (interview: 53.33%, survey: 58.31%), served in nursing departments (interview: 33.33%, survey: 37.62%), and had between 1 and 10 years of teaching experience (interview: 33.33%, survey: 47.64%). 

Of the online respondents, 65.8% (*n* = 210) reported that the school they work in has a policy for integrating technology into the curriculum and had integrated at least one technological method, and 70.22% responded that they had learned how to integrate technology into the curriculum. Of the qualitative interviewees, only 40% mentioned that their school had policies for integrating technology into the curriculum, and only three (20%) reported that they learned technology integration in their courses.

### 3.2. Considerations of Technology Integration in Curriculum Design

Five main themes regarding the considerations of technology integration in curriculum design were identified from 968 meaning sentences. Those themes were that integrating technology (i) supplements traditional teaching methods, (ii) heightens the immediate educator–student interaction, (iii) helps track the learning process and assists in record keeping, (iv) helps keep up with technological trends, and (v) advances professional learning objectives according to varying student backgrounds (see Table 2).

#### 3.2.1. Supplementing Traditional Teaching Methods

Technological assistance removes the temporal and spatial limitations that are generally part of traditional learning methods as certain technologies (i.e., pre-recorded video sessions) allow learning to take place anytime and anywhere, as often as desired. The need to be able to review and repeat lessons on complicated health-related subjects on demand was a motivating element, expressed predominantly by the educators in nursing, medicine, and pharmacy, for integrating technology in curriculum. 

I encourage my students when they don’t understand the material (such as biological mechanism and bio-statistic[al] analysis) in class … [to] go back and review the material through the learning platform. Moreover, if there [is] new information related [to] the subject, I could upload it anytime and anywhere. That is so great (an educator in a medical department, P-1012-44).

#### 3.2.2. Heightening the Immediate Educator–Student Interaction

Many curricula related to the health professions incorporate laboratory classes (labs). It is through these labs that educators can observe the students’ learning progress to confirm whether the students have achieved the learning goals. These labs enable educators to interact with students and immediately discuss subsequent steps with the students in the classroom. These outcomes could not be achieved without technical assistance. Technology permits educators to adjust teaching strategies according to students’ responses at any time in the classroom, through interactive response systems (IRSs) or social media, thus providing real-time feedback to understand how students process and react to the content. 

During the course, I play a video to allow students to experience the urgency of emergency medicine and to initiate discussion. If my students have any questions, I can immediately post literature on LINE (an instant messaging app) to respond [to] their question and allow them to read immediately and express feedback. What responses I need to give depends on the reaction of the students, and I can make adjustments at any time (an educator in a department of hospital management, 1003-11).

#### 3.2.3. Tracking the Learning Process and Record Keeping

Some schools have established an e-learning platform that requires educators to upload course materials that students can access any time. On this platform, complete student records can be stored in the cloud, assessments and examinations can be conducted without pen and paper, and medical students can access materials at their convenience. In addition, educators can monitor and track students’ record synchronously.

Now when my students attend class, lecture quizzes and practice are both web-based, and I no longer have to keep paper records of students’ grades. In addition, when assessing students, I no longer have to look for their homework in paper. Since everything is on the e-platform, it is complete and safely stored (an educator in a department of radiology oncology, 1004-81).

#### 3.2.4. Keeping up with Technological Trends

Educators realize that technology is a modern trend. Furthermore, educational technology is not the only aspect of our lives that is becoming digitized, as various other aspects of our lives are also deeply connected to technology. When educators see students constantly engaged with technological products, they fear that if they do not keep up with technological trends, their educational methods will become irrelevant and ineffective. 

Every student has a smartphone in hand, knows how to surf the web, and knows how to use apps. If our teaching can’t keep up, we will become obsolete (an educator in a nursing department, 1006-18).

#### 3.2.5. Advancing Professional Learning Objectives According to Different Student Backgrounds

During the interviews, the educators commented that, at any given level, the content that students are expected to learn as presented in the classroom is different from what the students will experience in clinical practice. In order to simultaneously deliver the content knowledge required for professional development and meet each student’s particular learning needs, the educators attempt to integrate technology into the curriculum to enhance the effectiveness of the educational experience. An educator teaching pediatrics nursing stated:

You know that two-year University of Technology students were licensed staff and had care experience in paediatrics, gained from 5 years of junior college. They ([the] students) could not [be] satisfied by attending a classroom-oriented course alone. I try to provide the opportunity to them [to] experience[e] actual cases in paediatrics. So, I use 3D avatars (three-dimensional animation) with high-fidelity and hands-on learning activities to allow them to practice repeatedly and formulate a process for each skill (an educator in a nursing department, 1001-35).

### 3.3. The Facilitators of Technology Integration in Curriculum

#### 3.3.1. Easy-to-Use E-Learning Platform Established by an Institution

More and more universities are emphasizing technology integration in their curriculum. One common method for technology integration is by establishing university-wide e-learning platforms. After training educators on how to use these convenient and simple e-learning platforms, educators are encouraged to integrate technology into their specific curriculum.

Our university uses the XXX platform, which allows us to store course data, exams, and homework. This way, all course material is electronic, and I don’t have to look for an appropriate platform myself. It’s easy for us to get started, and eventually we get used to using technology to store course materials on the platform (an educator in a pharmacy department, 1003-67).

#### 3.3.2. Inspired by the Positive Experience of Technology Integration from Colleague Team Interaction

Health profession education requires that a team of educators hold professional discussions to prepare appropriate coursework. If an educator on the team is enthusiastic about using educational technology, his/her experience could be an influencing factor in helping to motivate educators who are less enthusiastic about using the technology. This is especially the case if the experienced educator explains the process or method clearly. This positive experience may inspire other educators with less technological experience to try to make an effort to integrate technology into the curriculum.

At the beginning, I didn’t want to use it, but one of the paediatrics instructors in our group was very interested in using technology. Because we had to prepare units together, we often heard him talk about how to use technology, and it seem[ed] not difficult. So, I also want to say that we should be using it (an educator in a nursing department, 1003-11).

#### 3.3.3. Expectations from Students Who Have Used Technology Integration in Other Curricula

Students who have completed technologically-integrated courses may specifically ask educators whether technology integration can be used. Once aware of these students’ desires or expectations, educators may be more apt to consider integrating technology in the course.

While taking courses with certain instructors, students use gamified methods and accumulate points. Later, [when] they participated in my course, … some students ask[ed] me whether there will be opportunities for similar activities, which makes me think about trying it as well (an educator in a medical department, 1031-37).

### 3.4. The Barriers of Technology Integration in the Curriculum

#### 3.4.1. Inevitability of Technical Problems During Class

If coursework is accessible only on these technology platforms, problems with participation may arise when these platforms malfunction or when internet connections are faulty. Given the inevitability of technical problems, educators must be capable of managing problems with the technological devices and platforms; for example, when the students are unable to download apps, open files, or log in to Wi-Fi.

Some students have problems with the smartphone app, or [sometimes] it is incompatible with their phone. I have to stop and help them solve these problems; otherwise, they can’t move forward (an educator in a nutrition department, 2045-89).

#### 3.4.2. Need to Invest Extra Time to Prepare for the Course

Creating content for technology integration, such as taking pictures, editing, creating voiceovers, streaming, or designing apps, is more time consuming than creating traditional content. In addition, content must be updated annually based on the students’ academic or experiential backgrounds and characteristics. Although TEL courses appear convenient and contain educational content that can be rapidly changed, it is rather time-consuming to prepare the content.

If I start from photographs, the whole process of curriculum design to course delivery and including editing is very time consuming. If I use already available images, I still have to edit them before they can be added as images in the course, which is very time consuming (an educator in a nursing department, 2012-35).

#### 3.4.3. Concern that Students are More Interested in Using the Technology than Learning the Content

The use of technology enhances students’ interest; however, one complication educators encounter when designing a curriculum is figuring out how to integrate technology such that students not only enjoy learning but that they are also motivated to continue the learning process. 

Using the instant response system or playing a video clip in class can keep students concentrating on learning. However, students always play [on] their cell phone once … these tools [are no longer in use]. Since I start using technology in my teaching, I keep asking myself: “Do I use too much technology to make students only notice the technology?”, “Are there any alternative ways to enhance students’ motivation?” (an educator in a nursing department, 1035-67)

### 3.5. TPCK Scores and Factors Influencing Technology Integration

The mean score of the 32-item TPCK instrument for the sample was 176.82 (*SD* = 22.22), where the total score for a participant is the sum of their Likert scale responses for the 32 items. In terms of dimensions, the participants performed better in content knowledge (CK) (*M* = 6.47, *SD* = 0.53) and technological content knowledge (TCK) (*M* = 5.86, *SD* = 0.93), and poorer in technological knowledge (TK) (*M* = 5.00, *SD* = 1.09) and technological pedagogical content knowledge (TPCK) (*M* = 5.07, *SD* = 1.19) (see Table 3).

Table 4 shows the hierarchical multivariable logistic regression analysis, which was conducted to analyze the associations between genders, academic degrees, learned technology in teaching, TPCK scores, and school policies of the study participants. 

Gender and academic degree were entered into the analysis as model 1 and explained 8.8% of the variance. For model 2, which controlled for gender and academic degree, prior technological knowledge in teaching significantly influenced technology integration in curriculum and explained an additional 4.8% of the variance. Model 3 was adopted as the final model and explained a total of 73.6% of the variance to characterize the strong and significant influences of TPCK (OR = 3.50, 95% CI = 2.07–5.94) and school policy (OR = 4.59, 95% CI = 1.45–4.59) for technology integration in curriculum, which explained an additional 60% of the variance whereas the OR of learned technology (OR = 1.83, 95% CI = 1.06–3.16) in teaching and academic degree decreased. Consequently, an HPE who possessed a bachelor’s or master’s degree, has learned technology in teaching, has higher scores in TPCK, and teaches in a school with an technology integration policy that was more likely to integrate technological methods in their curriculum.

## 4. Discussion

### 4.1. Considerations of Technology Integration among Health Profession Educators

It is our belief that this survey provides insights into TPCK scores and the variables affecting technological integration that, once understood, can be used as evidence to make informed decisions to help in the planning of educational activities for HPEs. 

The results of this study were similar to the results of a systematic review conducted by Kirkwood and Price [32]. Our study indicated that the use of technology integration as a supplement to traditional teaching methods was the most common reason that HPEs integrated technology into their curriculum, while the Kirkwood and Price study revealed that 50% of technology integration interventions were used as supplementary activities in each curriculum. When educators initiate technology use for educational purposes, they tend to use it for education-centered activities to support traditional instructional applications. HPEs believed that technology integration might provide additional support and opportunities to help students learn under the constraints of traditional teaching. Moreover, with respect to integrating technology into a curriculum, the HPEs in the present study were interested in technology integration in order to enhance student motivation and encourage immediate interaction with students; these interests were consistent with those in previous studies [33].

An important consideration for technology integration in our study was adapting the professional learning objectives according to students’ backgrounds, indicating a learner-centered objective in curriculum design. Similarly, Gaikwad and Tankhiwale [34] mentioned that educators believe the convenience and flexibility of a technologically integrated curriculum allows them to adjust their teaching strategies depending on student input and participation. In this respect, educators could integrate technological methods to carry out effective instruction by considering multiple variables, such as student characteristics, subject area, classroom environment, and so forth that would be particularly helpful in the complex field of healthcare in which the health profession students must be able to face actual clinical situations. Thus, the ability of technology to simulate clinical situations and permit interactive decision-making fulfils educators’ curriculum design goals [3,35]. 

Given that the use of mobile devices has already become nearly indispensable in daily life, the ubiquity of technology has sparked an unavoidable trend for educators. Accordingly, the technologically altered learning needs of students have become important factors driving technology integration by educators [35]. In the present study, the ubiquity of technology and students’ expectations are perceived as driving forces for technology integration in curriculum. 

### 4.2. The Facilitating and Hindering Factors of Technology Integration in Curriculum Among Educators

The factor hindering technology integration most commonly mentioned by educators is time [36]. Time is required to produce material for technologically integrated courses; this is particularly the case for educators who are not familiar with the technology, which could, therefore, cause stress [37]. In the past, organizational support, which included training and the provision of technical resources and time, was an influencing factor related to whether educators would use a technologically integrated curriculum [16,37]. As many studies have revealed, insufficient technical support, which caused educators to be less confident about technology integration, was also an impeding factor [38]. This information from prior studies supports the results of the present study because a convenient learning platform and training for those platforms already existed at the universities, which increased the motivation to use technology. 

### 4.3. The Influencing Factors of TPACK in Considering Technology Integration

Consistent with the findings of an earlier study, gender is an influencing factor of educator technology integration [4]. The suspected reason for this gender-based difference is that men are more accepting of technology than women. Therefore, male educators were more likely to integrate technology in a curriculum than female educators [39]. 

With respect to the educator’s academic degree or education preparation level, prior studies have produced inconsistent findings. However, it is reasonable to assume that higher education, especially at the doctoral level of education for health professionals, focuses on the advancement of professional knowledge and theoretical development rather than strengthening its technological knowledge. HPEs in universities are experts who specialize in health-related subject matter and, ergo, may not have had any pedagogical or technological training before they became educators. Moreover, one of the responsibilities for educators in higher education is to participate in grant programs or studies. In contrast, bachelor’s and master’s level educators may be required to become more instruction-oriented, shifting a greater focus toward the integration of technology into the curriculum.

The number of years an educator has spent teaching does not affect the educator’s technology integration [17]. Instead, as we determined in our study, and as supported by the results of Koh et al. [40], the key factor for the successful integration of technology into a curriculum was whether or not educators had participated in technology training. However, when TPCK and school policies exist, gender influence disappears and the impact of learned technology also decreases, indicating that TPCK and school policy are the strongest factors affecting educator’s technology integration. Undoubtedly, TPCK is the most important variable with regard to educators’ technology integration, in our study as well as in previous studies [4,41]. Notably, in our findings, school policies that provide technological support and equipment were not only facilitators of integration, identified in the qualitative interviews, but also the strongest factor for technology integration based on the results of the online survey (see Hew and Tan [42]).

Before coming to a close, we will discuss the response rate. From our nation-wide online sample, 32.3% HPEs submitted their online questionnaires. Although the response rate does not seem high, compared with the previous online questionnaire survey, this rate is acceptable [29].

## 5. Conclusions

Due to the impact of technology on life and learning, educators must consider the maximum learning benefits of technological integration between student learning preferences and learning outcomes. We explored the considerations and the influencing factors for the technological integration of an educational curriculum based on the technological pedagogical content knowledge (TPCK) framework for HPEs. A mixed methodology was used, including semi-structured interviews and an online TPCK survey with a randomly selected sample from Taiwanese universities. Five themes emerged; namely, supplementing traditional teaching methods, improving immediate educator–student interaction, tracking the learning process and improving the record keeping, keeping up with technological trends, and advancing professional learning objectives for different student needs. The results found that gender was an influencing factor of educator technology integration. However, when TPCK and school policies existed, gender influence disappeared and the impact of prior technological knowledge also decreased, indicating that TPCK and school policy were the strongest factors affecting educator’s technology integration. To improve the use and effectiveness of technologically integrated curriculum, HPEs’ in-service programs could focus on technology integration training to enhance educators’ TPCK. The risk of technical issues and the need to invest extra time in the preparation process are obstacles to technical integration for authors. Therefore, schools can provide educators with relevant resources and support, and encourage them to interact with colleagues to enhance their confidence in technology integration. Although TPCK is a specialized, highly applied type of knowledge that supports content-based technology integration, it still has limitations, such as the inability to describe the more fluid processes that happen as educators work with students. 

## Figures and Tables

**Figure 1 ijerph-16-02602-f001:**
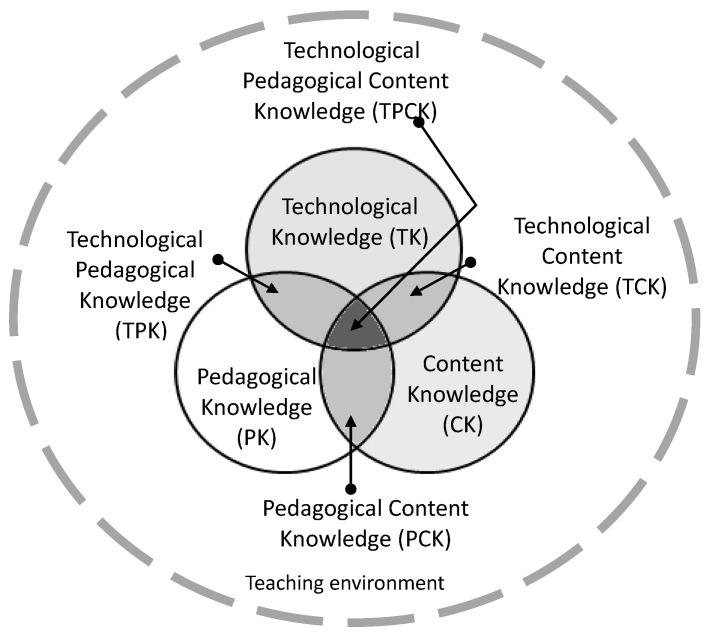
The TPCK framework (Mishra and Koehler, 2006).

**Table 1 ijerph-16-02602-t001:** Descriptive characteristics of participants.

Variables	Interview (*n* = 15)		Survey (*n* = 319)	
	N	%	N	%
**Gender**				
Male	6	40.00	117	36.68
Female	9	60.00	202	63.32
**Age**				
21–30	0	0	2	0.63
31–40	1	6.67	46	14.42
41–50	9	60.00	156	48.90
≥ 51	5	33.33	115	36.05
**Academic degree**				
Bachelor	0	0	26	8.15
Master’s	7	46.67	107	33.54
Doctorate	8	53.33	186	58.31
**Profession**				
Nursing	5	33.33	120	37.62
Radiation oncology	1	6.67	5	1.57
Elderly care	0	0.00	37	11.60
Nutrition	1	6.67	21	6.58
Physician	4	26.67	64	20.06
Medical examination	2	13.33	58	18.18
Pharmacy	2	13.33	14	4.39
**Total years of teaching experience**				
1–10 years	5	33.33	152	47.64
11–20 years	7	46.67	100	31.35
21 years or more	3	20.00	67	21.01
**School policy**				
Yes	6	40.00	210	65.8
No	9	60.00	109	34.2
**No. of technology integrations**				
0	0	0	109	34.2
1	9	60.00	122	38.2
>1	6	40.00	88	27.6
**Learned technology in teaching**				
Yes	3	20.00	224	70.22
No	12	80.00	95	29.78

**Table 2 ijerph-16-02602-t002:** Qualitative analysis of health profession educators (N = 15).

Theme (Abbreviations)	Categories	Definition
1. Supplementary for traditional teaching (S)	■ Absent learning in the classroom ■ Repeat the study of teaching materials at any time ■ Various learning resources in the form of non-textbooks	Technological assistance allows learning to not be limited by time or location and can be repeated at any time.
2. Heightening the immediate educator–student interaction (H)	■ Use of IRS to interact with students ■ Exams with feedback immediately after the answer is given ■ Posing questions at any time in the classroom on social software	Through instant interaction and understanding of the process and reactions of students during the course, teaching and learning strategies can be adjusted at any time.
3. Tracking the learning process and record keeping (T)	■ Easy to keep an e-profile of students ■ Know the questions of students in previous years ■ All students’ assignments and examinations are scored on the e-platform through online course activities.	In e-learning, complete student records can be stored in the cloud, assessments do not require pen and paper, and medical students can review materials at any time when studying. In addition, educators can monitor their students synchronously.
4. Keeping up with technological trends (K)	■ The needs of technology use everywhere ■ Falling behind in the age of technology ■ Using technology is modern	Technology is a modern trend. Not only is educational technology becoming digitized, but various aspects of real life also require the use of technology.
5. Advancing the professional learning objectives according to different background of students (A)	■ Professional training requires advanced situations ■ Provide a holistic and advancing clinical situation to learn	The integration of student needs into TEL is used as a pedagogical method to achieve professional knowledge of content.

Abbreviations: IRS: Interactive Response System; TEL: Technology-Enhanced Learning.

**Table 3 ijerph-16-02602-t003:** Summary of description results of TPCK.

	No. of Item	Cronbach α	Range	Mean	Standard Deviation
CK	3	0.91	13–21	6.47	0.53
PK	6	0.91	15–42	5.68	0.72
PCK	3	0.86	7–21	5.40	0.90
TK	7	0.7	10–49	5.00	1.09
TPK	5	0.83	10–35	5.83	0.89
TCK	3	0.92	6–21	5.86	0.93
TPCK	5	0.81	7–35	5.07	1.19

**Table 4 ijerph-16-02602-t004:** Multivariable logistic regression model for technology integration in the curriculum.

Variable	No (%)	Yes (%)	Model 1	Model 2	Model 3
			OR (95% CI)	OR (95% CI)	OR (95% CI)
Gender					
Female	55(46.2)	147(72.8)	1	1	1
Male	54(46.2)	63(53.8)	2.08(1.28–3.40) **	1.83(1.11–3.03) *	1.65(0.98–2.79)
Academic Degree					
Bachelor’s and Master’s	30(23.1)	100(76.9)	1	1	1
Doctoral	78(41.7)	109(58.3)	0.47(0.28–0.78) **	0.45(0.27–0.76) **	0.40(0.23–0.69) **
Teaching Experience					
<10	41(39.8)	62(60.2)		1	1
≥10	68(31.5)	148(68.5)		1.26(0.75–2.12)	1.17(0.68–2.02)
Learned Technology in Teaching					
No	48(56.5)	47(49.5)		1	1
Yes	61(27.2)	163(72.8)		2.38(1.42–4.01) ***	1.83(1.06–3.16) *
TPCK					
<mean scores	81(40.5)	119(59.5)			1
≥mean scores	28(23.5)	91(76.5)			3.50(2.07–5.94) ***
School Policy					
No	132(83.0)	27(170)			1
Yes	15(9.4)	145(90.6)			4.59(1.45–4.59) ***
Variance explained			8.8%	13.6%	73.6%
−2 log likelihood			358.92	373.87	383.66
df			3	5	7
△X^2^(△df)				5.6(1)	65.4(3)
P				0.002	<0.001

Abbreviations: OR, odds ratio; 95% CI, 95% confidence interval; df, degree of freedom.

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
