# Peer review of "Factors Influencing Technology Integration in the Curriculum for Taiwanese Health Profession Educators: A Mixed-Methods Study"

_ijerph, 2019, doi:10.3390/ijerph16142602_

Round 1

Reviewer 1 Report

I have passed through the paper a couple of times. I annotated a lot of changes on the pdf, both typographic and in terms of layout. Overall, the paper is quite straightforward and the ideas are reasonably well expressed. Some care is needed with the English language and some more work is required to give the paper a better flow. The results are to be as expected I think but the authors could have included some future work. I know too that some effort has been made to considered the results from the perspective of the health professionals they have been examining. I'm not sure if anything unique comes across about health professionals that would not apply to any other group. However, the questions in the first place are more general than referring to specific medical uses because it considers professional across the complete working spectrum. The paper does need some work to raise its level but I think it could be done in a reasonably short time frame.

Author Response

We express our thanks for your recognition of the results of this study. We hope this study can provide a reference for on-the-job education training of technological integration in curriculum for health professional teachers.

Reviewer 2 Report

The submission by Chen et al. is highly topical and addresses vital questions that influence technology integration in health education curricula. The mixed-methods study was designed and executed immaculately and the results from the study provide a good foundation for actionable recommendations. The manuscript is well-grounded in literature and provides an authoritative and insightful analysis of the data. The inferences derived by the authors are well-supported by the data. It was not surprising to see that institutional support and policies regarding technology usage and integration are among the most important drivers that affect curriculum decisions. I hope that the manuscript provides supporting evidence for Departments and colleagues that petition administrative divisions for technological support and upgrades to improve educational outcomes. I commend the work carried out by the authors.

Author Response

1. We did not see the PDF that you annotated a lot of changes on.

2. Thank you for the comment. The revised manuscript has been reviewed and edited by an English editing expert.

3. Thank you for the comment. This article is the first study to analyze TPACK for health professional educators and the preliminary results did not address the application of special health professional teaching. In the future, we should be able to analyze the differences in specific health professionals and its applications.

Reviewer 3 Report

Reviewer comments are given in the attached file.

Round 2

Reviewer 3 Report

Thank you for your precise and systematic response to my review. I have one last issue that needs to be addressed in the manuscript. You need to briefly discuss the different purposes for

qualitative and quantitive research, as part of your mixed-methods approach. What does each of these methodologies contribute to your research? What are the different purposes?

This might be best done at  2. Materials and Methods.

Author Response

Thank you for the comment. We agree with you mentioned to add the  different purposes for qualitative and quantities research on page 2-3,line 93-99.

The focus of qualitative research is on the process, that is, how exactly TPCK exited. In the development of TPCK process, how the HPEs’ considers are in the process, what their attitudes are, how they have changed, what problems they have encountered, what they are confused about, what their feedback is and what their final results are, etc. Since there is no research on HPEs’ TPCK, this mixed research was design with qualitative research and then quantitative research. It can help to clarify whether HPEs have TPCK and its’ influencing factors.